# HDL Functions—Current Status and Future Perspectives

**DOI:** 10.3390/biom13010105

**Published:** 2023-01-04

**Authors:** Yasuhiro Endo, Masanori Fujita, Katsunori Ikewaki

**Affiliations:** 1Division of Anti-Aging and Vascular Medicine, Department of Internal Medicine, National Defense Medical College, Saitama 359-8513, Japan; 2Division of Environmental Medicine, National Defense Medical College Research Institute, Saitama 359-8513, Japan

**Keywords:** HDL, cholesterol efflux capacity, reverse cholesterol transport

## Abstract

Cardiovascular disease (CVD) is the leading cause of death in Western countries. A low HDL-C is associated with the development of CVD. However, recent epidemiology studies have shown U-shaped curves between HDL-C and CVD mortality, with paradoxically increased CVD mortality in patients with extremely high HDL-C levels. Furthermore, HDL-C raising therapy using nicotinic acids or CETP inhibitors mostly failed to reduce CVD events. Based on this background, HDL functions rather than HDL-C could be a novel biomarker; research on the clinical utility of HDL functionality is ongoing. In this review, we summarize the current status of HDL functions and their future perspectives from the findings of basic research and clinical trials.

## 1. Introduction

Cardiovascular disease (CVD) is the primary cause of death in Western countries and Japan. In the 1970s, the Framingham Heart Study showed that HDL-C was negatively associated with the development of cardiovascular disease (CVD) [1]. Accordingly, clinicians regarded HDL as anti-atherogenic lipoprotein and expected HDL-C-raising therapy to reduce CVD events. Unfortunately, clinical trials using niacin and cholesterol ester transfer protein (CETP) inhibitors mostly failed to reduce CVD events [2,3,4,5]. Conversely, prospective studies showed that cholesterol efflux capacity (CEC), a principal anti-atherogenic function of HDL, is a negative predictor of CVD, bringing about a paradigm shift from HDL quantity (HDL-C) to functions.

Sixty years ago, Glomset proposed the concept of reverse cholesterol transport (RCT) for HDL [6], and since then, HDL research has gained momentum in both basic and clinical studies. In this review, we summarize the current views in HDL research, as well as future perspectives on HDL functions.

## 2. HDL-C and Atherosclerosis

The Framingham Heart Study, as noted above, and the PROCAM study, have shown that HDL-C is negatively associated with CVD risk [1,7]. Thus, high HDL-C has been considered to be atheroprotective for a long time. However, recent prospective cohort studies in Denmark found an increased hazard ratio for all-cause mortality in individuals with extremely high HDL-C [8]. In addition, pooled analysis (EPOCH-JAPAN: Evidence for Cardiovascular Prevention from Observational Cohorts in Japan), including nine cohort studies in Japan, confirmed the above finding [9].

Using genome-wide association analysis (GWAS), Zanoni et al. reported that a loss-of-function mutation of scavenger receptor B1 (SR-B1) increased the risk of coronary heart disease (CHD) despite elevated HDL-C [10]. Furthermore, CETP inhibitors, potent HDL-C-raising drugs, failed to show a protective effect against the development of CVD [2,4,5]. These findings, therefore, led to growing recognition that HDL functions are more important than HDL-C levels.

## 3. HDL Functions

Traditionally, researchers have focused on the size of HDL and the proteins and phospholipids it contains, classifying them by density gradient ultracentrifugation, two-dimensional gel electrophoresis, nuclear magnetic resonance spectroscopy, or ion mobility [11]. In addition, recent technologies, including proteomics and lipidomics, have uncovered the heterogeneity and complexities of HDL, that they harbor various apolipoproteins, lipid transfer proteins, enzymes, and phospholipids [12,13]. This has led to the discovery that HDL has several anti-atherosclerotic functions, including reverse cholesterol transport (RCT) and anti-inflammatory, antioxidative, vasodilatory, and anti-thrombotic functions [14,15,16,17,18,19].

In this study, we have summarized HDL functions from the aspects of basic and clinical research.

### 3.1. Reverse Cholesterol Transport

As shown in Figure 1, RCT is believed to have a major anti-atherosclerotic function. RCT is a pathway by which HDL transports excess cholesterol from foamy macrophages and smooth muscle cells, and peripheral tissues to the liver [20]. Cholesterol transported to the liver by HDL is partly stored as cholesteryl esters, while the rest is excreted as bile. RCT, as well as trans-intestinal excretion (TICE) [21], play pivotal roles in cholesterol efflux into the intestinal tract.

Apolipoprotein A-I (ApoA-I), a major apolipoprotein of HDL, is mainly secreted from the liver and small intestine. The secreted apo A-I gains phospholipids and cholesterol via ATP binding cassette transporter A1 (ABCA1) from peripheral tissues to form discoidal HDL (preβ HDL) and further effluxed cellular cholesterol via ATP binding cassette transporter G1 (ABCG1), and SR-BI, which leads to the formation of mature HDL particles by lecithin cholesterol acyltransferase (LCAT) [22,23,24,25]. The majority of cholesterol ester (HDL-CE) is transferred from HDL to VLDL and LDL by CETP and is taken up by LDL receptor (LDLR) in the liver [26].

Many studies have revealed that enhancing RCT promoted atherosclerosis regression, and that there were two ways of achieving this—through improvements in macrophage cholesterol efflux, and of hepatic uptake and biliary excretion [20,27]. In these approaches, the initial cholesterol efflux from macrophages is the primary determinant of RCT [28,29]. As indicated above, in the first step, lipid-poor ApoA-1 acts as an acceptor in ABCA1-mediated efflux, while ABCG1 and SR-B1-mediated efflux is mediated by mature HDL. Our laboratory previously focused on macrophage cholesterol efflux and found that overexpression of stearoyl-coenzyme A desaturase 1 (SCD1) and proteasomal inhibition of ATP-binding cassette transporter A1 (ABCA1) and ABCG1 in macrophages promoted RCT in vivo [30,31]. Meanwhile, Rothblat and Rader’s group had established an assay for cholesterol efflux capacity (CEC) [32] to assess it from macrophages in clinical research. In their assay, ABDS (ApoB-depleted serum) was added to the cholesterol-radiolabeled cultured cells (3H-Cholesterol). After incubation, the medium was collected, and the lipids in the cells were extracted using hexane and isopropanol. The [3H] radioactivity of the medium and cell extracts was measured, and the CEC was calculated by dividing the medium count by the sum of the medium and cell counts. Therefore, CEC reflects the ability of the patient’s HDL to expel radiolabeled cholesterol from the cultured cells.

Cross-sectional and prospective studies revealed that CEC was a negative predictor of cardiovascular disease development independent of HDL-C in the Dallas Heart Study and EPIC-Norfolk study [33,34,35]. In line with these findings, we previously reported that CEC was a predictor of coronary artery disease (CAD) in Japanese patients [36]. In addition, the association of CEC with CVD was confirmed in a meta-analysis, making it irrefutable that CEC is a negative indicator for CVD. However, there are some limitations on CEC assays, and they remain to be standardized because the protocol for CEC requires tracers and cultured cells. To overcome these difficulties, a cell-free cholesterol efflux assay (Cholesterol Uptake assay, CUC) has been established in Japan [37,38,39].

In CUC, measurements are made automatically using biotin-PEG-labeled cholesterol and anti-ApoA-1 antibodies in a tracer- and cell-free assay. Briefly, biotin-labeled PEG-cholesterol was added to the patients’ ABDS. The HDL was captured using an anti-ApoA-1 mouse monoclonal antibody. After enzymatic labeling with streptavidin, a chemiluminescent substrate was added, fluorescence intensity was measured using a spectrophotometer, and CUC was measured. Therefore, CUC reflects the HDL’s ability to bind cholesterol.

In clinical trials, CUC was found to be negatively correlated with plaque burden (lipid index) in an optical coherence tomography (OCT) study and associated with subsequent revascularization after percutaneous coronary intervention (PCI) [37,38,39]. CUC is a practical alternative to CEC and a promising predictor of future CVD (Table 1).

### 3.2. Anti-Inflammatory Functions

Inflammation aggravates atherosclerosis progression, coordinated with a variety of immune cells and inflammatory signaling pathways, including Toll-like receptor (TLR) signaling and inflammasomes [43]. The CANTOS and COLCOT trials provided proof for the pivotal role of inflammation in atherosclerosis [44,45].

Monocyte recruitment is known to be a first step in atherosclerosis formation, and HDL has anti-inflammatory functions that target endothelial cells. In this regard, Cockerill et al. reported that HDL exerted anti-inflammatory effects via reducing cytokine-induced adhesion molecules, such as VCAM1 (vascular cell adhesion molecule-1) [46]. In addition, Tabet found that HDL regulated ICAM-1 (intracellular adhesion molecule-1) via HDL-transferred microRNA-223 [47]. Furthermore, Scanu observed that HDL attenuated monocyte activation by inhibiting MCP-1 secretion in endothelial cells [48].

As for research other than on endothelial cells, Murphy et al. focused on anti-inflammatory effects on immune cells and reported that HDL and ApoA-1 attenuated neutrophil activation in vitro and in vivo [49]. Additionally, De Nardo reported that HDL reduced the expression of TLR-induced cytokines in macrophages by regulating the transcription factor ATF3 [50], and Wang et al. observed that HDL and ApoA-1 attenuated T cell activation via disruption of lipid rafts [51]. Regarding inflammasomes, which were mentioned at the beginning of this section, Thacker showed that HDL also had the ability to inhibit inflammasome activation induced by cholesterol crystals [52].

From another point of view, ABCA1- and ABCG1-mediated cholesterol efflux due to HDL was reported to be modulated by the immune cell response. In particular, previous studies revealed that myeloid ABCA1/ABCG1 deficiency activated myeloid cell proliferation and Nrp3 inflammasomes [53,54], and T cell ABCA1/ABCG1 deficiency induced T cell activation and senescence [55]. In summary, HDL exerts anti-inflammatory functions by affecting several factors in the formation of inflammation in vitro and in vivo.

Regarding use in clinical settings, Rader’s group evaluated the ability to inhibit LDL oxidation in terms of a high-density lipoprotein inflammatory index (HII), and applied it in the JUPITER (Justification for the Use of Statins in Prevention: an Intervention Trial Evaluating Rosuvastatin) sub-analysis. They found that the HII was associated with the incidence of CVD [41]. Other researchers had focused on the inhibitory effect of tumor necrosis factor-α (TNF-α)-induced VCAM1 mRNA expression by HDL, known as the HDL anti-inflammatory function [56], and noted that anti-inflammatory capacity was inversely associated with CHD risk in the PREVEND (Prevention of Renal and Vascular End Stage Disease) study [42] (Table 1).

### 3.3. Anti-Oxidative Functions

The anti-oxidative function of HDL is to remove lipid hydroperoxide (LOOH) and oxidized phospholipids (oxPL) from atherogenic oxidized low-density lipoprotein (oxLDL) [57]. Previous articles in this regard focused on HDL-associated hydrolases with an inhibitory effect of LDL oxidation. For example, paraoxonase (paraoxonase1:PON1), bound to ApoA-1, reduced lipid peroxidation [58], and mice lacking paraoxonase were susceptible to LDL oxidation and atherosclerosis [59].

In clinical studies, a PON1 genotype (PON1 QQ192) with decreased serum PON1 activity was associated with cardiovascular risk [60]. A platelet-activating factor acetyl hydrolase (PAF-AH), also known as lipoprotein-associated phospholipase A2 (Lp-PLA2), had the ability to hydrolyze phosphatidylcholine hydroperoxide (PLCOOH) and ox-PL [61,62]. In addition to these proteins, lecithin-cholesterol acyltransferase (LCAT), an enzyme for cholesterol esterification, had the ability to hydrolyze ox-PL [63].

Garner showed that methionine residues of apolipoprotein AI and AII played a pivotal role in the reduction in LOOH [64]. Davidson et al. focused on the association of HDL composition with anti-oxidative functions and found that apoD, apoF, apoJ, apoL1, and apo M were associated with LDL oxidation [65]. In summary, HDL exerted anti-oxidative functions in coordination with several apolipoproteins and HDL-associated hydrolases.

### 3.4. Vasodilatory and Antithrombotic Functions

Endothelial dysfunction is mainly involved in early atherosclerosis progression by disrupting homeostasis of nitric oxide (NO) production [66]. HDL regulates endothelial NO synthase (eNOS), thereby modulating endothelial homeostasis. In 2001, Yuhanna first reported that HDL upregulated eNOS in endothelial cells via scavenger receptor B1 (SR-B1) [67]. Then, stimulated by HDL, eNOS was reported to be regulated by Akt, mitogen-activated protein (MAP) kinase, lysophospholipid receptor (S1P3), and the PDZ domain of SR-B1 in the endothelial cell [68,69,70].

From another point of view, Besler showed that HDL isolated from patients with CAD, including acute coronary syndrome (ACS), reduced NO production via the inhibition of eNOS phosphorylation [71]. Platelet and coagulation cascades also play an essential role in the formation of thrombosis and atherosclerosis lesions. In addition to vasodilatory functions, HDL has anti-thrombotic functions, which modulate platelet activity and coagulation cascades by affecting Protein C and Protein S [72].

### 3.5. Anti-Infectious Activity

In recent years, the clinical utility of HDL for sepsis has drawn much attention.

In 1993, Levine discovered that HDL was bound to lipopolysaccharide (endotoxin), present in the outer membrane of Gram-negative bacteria [73]. Guo et al. found that ApoA-1 knockout in mice reduced the survival rate from sepsis, which was remedied in an ApoA-1 transgenic mouse [74]. Trinder et al. found that a CETP inhibitor (anacetrapib) increased the survival rate in CLP (cecal ligation and puncture)-mouse models [75]. In a 2022 study, reconstituted HDL (ETC-642) exerted beneficial effects on sepsis by attenuating LPS, LTA (lipoteichoic acid), and TNF-α-induced inflammation [76]. Therefore, an apoA-1 mimetic or a CETP inhibitor could be a novel therapy for sepsis in the future.

## 4. HDL Components

HDL consists of several apolipoproteins (A-I, A-II, E, etc.) and lipid components such as cholesterol, phospholipids, and triglycerides (TG). Recent studies using lipidomics, proteomics, and transcriptomics have discovered the disease-associated heterogenicity of HDL. Recent omics analyses on HDL components are summarized below.

### 4.1. HDL Lipidome

The HDL lipidome reflects the biological activities of HDL. In particular, phosphatidylcholine and sphingomyelin, major phospholipids in HDL [12], have been observed to modulate cholesterol efflux from macrophages in vitro [77]. In addition, plasmalogen levels of HDL have been clinically associated with cholesterol efflux capacity in patients with chronic kidney diseases [78]. Furthermore, S1P binding with ApoM in HDL exerts anti-inflammatory functions by inhibiting cytokine-stimulated ICAM-1 and VCAM1 in endothelial cells [79].

### 4.2. HDL Proteome

Technological advances in proteomic analysis have uncovered the diversity of the HDL proteome. According to the HDL Proteome Watch 2021 database, the HDL proteome was altered in atherosclerotic disease, renal disease, cirrhosis, and COVID-19 [13,80,81,82,83,84]. Gordon et al. found that 16 proteins in the HDL proteome were altered and associated with CEC in patients with CAD [85,86]. Additionally, the HDL proteome caused inflammatory remodeling in patients with psoriasis, which was associated with a higher incidence of CVD [87].

In evaluating the direct causality of the HDL proteome with functionality, Vaisar et al. showed that endotoxin treatment increased SAA (serum amyloid A) protein levels in the HDL proteome and decreased CEC, which was abolished in SAA KO mice [88]. Their findings indicate that inflammatory remodeling of the HDL proteome has a direct impact on HDL functionality in vivo.

However, HDL proteome analysis has certain limitations. First, because the methodology of proteomics analysis on HDL is not standardized, the proteins detected have varied among institutions. Second, conventional proteomics does not evaluate protein modification, such as the oxidation of ApoA-1, which affected HDL CEC [89,90]. However, by combining lipidomics with proteomics data, more detailed HDL functional analysis can be performed [91].

### 4.3. HDL-MicroRNA

MicroRNA (miRNA) is single-strand RNA with 21–25 nucleotides that regulate post-transcriptional gene expression in eukaryotes. MicroRNA 33 (mir-33), the target for sterol-regulatory element-binding factor-2, regulates HDL biogenesis by regulating liver ABCA1 [92]. In 2011, Vickers et al. reported that microRNA was encapsulated in HDL, and the microRNA profile differed in familial hypercholesterolemia as compared with healthy controls [93]. They also showed that HDL-delivered microRNA was dependent on SR-B1 and affected recipient cells. Research on HDL-microRNA is ongoing and may lead to clinical applications.

## 5. Environmental Factors and HDL Function

Environmental factors, including diet and nutrition, are associated with the development of atherosclerosis. Recently, a high-fat diet was found to modulate immune cells and induce inflammation in vivo [94]. Therefore, dietary and nutritional interventions are seen as important in preventing atherosclerosis and are attracting attention. This section focuses on environmental factors and HDL function.

### 5.1. Nutrition and HDL Functionality

Diet and nutrition contribute to atherosclerosis progression or prevention by affecting cholesterol homeostasis [95]. Polyphenols are bitter, pigmented compounds found in most plants and have anti-oxidative effects that are expected to prevent atherosclerosis. Some polyphenols enhance HDL functions in vivo and in humans. Previously, we reported that caffeic acid and ferulic acid, the prominent phenolic acids of coffee, upregulated ABCG1 and SR-B1 expression, and coffee consumption enhanced HDL-mediated cholesterol efflux in an ex vivo study [96]. We then showed that ferulic acid increased whole-body (in vivo) RCT in mice. In another study, Hirata and one of our colleagues reported that xanthohumol, a prenyl flavonoid from the hop plant (*Humulus lupulus* L.), promoted in vivo RCT in hamsters [97]. Another flavonoid, Quercetin, contained in onions, apples, and strawberries, upregulated cholesterol efflux via increased ABCA1 expression in THP-1 macrophages [98]. In addition, polyphenol-rich (366 mg/kg) olive oil consumed for 3 weeks improved HDL functionality, including cholesterol efflux and oxidative status in humans [99]. Furthermore, consuming a Mediterranean diet in a 1-year intervention enhanced HDL function, including CEC, HDL antioxidant properties, and HDL vasodilatory capacity [100]. Other research groups have focused on carotenoids, plant-derived natural pigments with anti-oxidative and atheroprotective properties. McEneny found that lycopene-rich diets (224–350 mg lycopene/week) increased HDL-PON1 activity in a randomized controlled trial [101]. In the past, we reported that Astaxanthin increased ABCA1 and ABCG1 expression in macrophages [102]. In addition to polyphenols and carotenoids, monounsaturated and polyunsaturated fats in the diet can enhance cholesterol efflux capacity [103]. In this regard, Yang found that saury oil, a fish oil rich in LCMUFAs (omega-11 long-chain monounsaturated fatty acids), increased HDL-C and CEC in double-blind, cross-over clinical trials [104].

### 5.2. Exercise and HDL Functions

Regular physical activity has beneficial effects on cardiometabolic parameters, including reduced cardiovascular mortality risk [105], improved glucose control [106], and anti-inflammatory functions [107]. Regarding effects on HDL functions, Sarzynski et al. found that regular prolonged exercise (20 kcal/kg per week) upregulated cholesterol efflux in two randomized clinical trials [108,109]. Additionally, walk–run training enhanced HDL’s anti-oxidative function by increasing PON1 activity in patients with metabolic syndrome [110]. In addition, in combination with a Mediterranean diet, 12 weeks of exercise improved CEC by attenuating the MPO (Myeloperoxidase) oxidation of HDL. Furthermore, physical training, including high-intensity interval training upregulated HDL-mediated eNOS phosphorylation and PON1 activity in heart failure with preserved ejection fraction [111]. However, the threshold of exercise intensity that ameliorates HDL function remains unknown and further studies will be needed.

### 5.3. Cigarette Smoking, Air Pollution, and Chemicals

Smoking is a well-known risk factor for CVD, and passive smoking affects lipid profiles in children and females [112,113,114,115]. In addition, ambient air pollution promotes early atherosclerosis and oxidative stress [116]. As underlying mechanisms for their increased atherogenicity, smoking, and air pollution are reported to attenuate HDL functions, including CEC and anti-oxidation [117,118]. As expected, smoking cessation improves CEC and increases HDL anti-inflammatory function [119]. Recently, plastic-associated chemicals, including bisphenol A (BPA) and phthalates, have been the focus of research owing to their atherogenicity [120]. In particular, BPA is associated with atherosclerosis [121,122,123] and exerts proinflammatory effects [124]. Although it also downregulated ApoA-1 expression [125], the effect of BPA on HDL function remains to be elucidated, and further studies are warranted.

## 6. HDL Functions in Diseases

### 6.1. Cardiovascular Disease (CVD)

As noted earlier, growing evidence for HDL atheroprotective effects has emerged in recent years and clinical trials on CVD have focused on HDL functionality. Khera suggested that CEC is an important predictor of CVD; this has been a driver of clinical research on the utility of HDL functionality as a biomarker [33]. In a cross-sectional study, our group focused on the clinical utility of CEC in Japanese CVD patients [36] and Thakkar highlighted the clinical usefulness of HDL functionality in ACS [126]. Ogura reported an inverse association of CEC with CVD risk in Japanese patients with familial hypercholesterolemia [40]. Rohatgi had also provided proof of this concept in a prospective cohort study [34], and studies following it found that CEC was inversely associated with the future development of CVD [127,128]. Based on these findings, a meta-analysis concluded that CEC was a negative risk factor for CVD. Lipid-lowering drugs for the prevention of CVD affect HDL functionality. For instance, we reported that ezetimibe enhanced in vivo RCT in hamsters [129]. In addition, Khera reported that a peroxisome-proliferator-activated-α (PPAR-α) agonist increased CEC in a randomized control trial [130]. Going beyond conventional lipid-lowering drugs, ApoA-1 mimetics, which enhance cholesterol efflux, have been receiving attention regarding inhibition of atherosclerosis progression. AEGIS-Ⅱ, a phase 3 clinical trial on an ApoA-1 mimetic (CSL112), is ongoing, and it could be a promising novel treatment for reducing CVD.

### 6.2. Chronic Kidney Disease (CKD)

CKD is independently associated with increased CVD morbidity and mortality [131,132]. Previous studies highlighted impairment of HDL function in CKD. For instance, Moradi found impaired PON1 activity in end-stage renal disease (ESRD) [133], and RCT was impaired in HDL isolated from ESRD patients [134]. In addition to RCT, Holzer et al. focused on an altered HDL proteome in patients on hemodialysis and showed that HDL-mediated cholesterol efflux was reduced [135]. Although altered HDL function may translate into increased CVD in CKD, Chindhy did not find this to be the case.

Recently, genetic variants of ApoL-1, a key functional component of the trypanolytic factor of HDL, were reported to be associated with susceptibility to CKD in African Americans [136]. ApoL1 gene variants were also associated with hypertension-induced nephropathy, focal segmental glomerulosclerosis, and HIV-associated nephropathy [137,138]. Thus, further evidence will be needed to determine whether ApoL1 is a potential therapeutic target.

### 6.3. Diabetes Mellitus

Diabetes mellitus (DM) is characterized by low HDL-C dyslipidemia with elevated CVD morbidity and mortality [139]. There is ample of evidence that DM significantly modulates HDL functionality, such as by altering the phospholipid composition of HDL and affecting RCT [140,141]. Vries was the first to report reduced cellular cholesterol efflux in type 1 DM [142] and Morgantini found that type 2 DM impaired anti-inflammatory and anti-oxidative functions of HDL [143] In addition, He et al. showed that ABCA1-CEC of small HDL was impaired in type 2 DM and SERPINA1 (serpin family A member 1) levels in the HDL proteome were altered, which was highly correlated with ABCA1-CEC [144]. Regarding pancreatic β cell functions, anti-oxidative capacity and cellular cholesterol efflux were positively associated with HOMA-β, an estimate of β cell function in T2DM [145]. Furthermore, Yalcinkaya reported that HDL inhibited the apoptosis of pancreatic β cells due to endoplasmic reticulum stress (ER) [146]. Other researchers highlighted the effects of ApoA-1 on insulin synthesis and secretion, finding that ApoA-1 enhanced insulin secretion and Irs, Ins, and Pdx1 gene expression via the G-protein-cAMP-PKA-FoxO1 axis in β cells [147]. The anti-diabetic effects of HDL are promising, but further clinical studies are warranted.

### 6.4. Autoimmune Disease

Rheumatoid arthritis (RA) is a chronic inflammatory autoimmune disease accompanied by increasing CVD risk [148]. Some clinical studies have uncovered HDL dysfunctionality in RA. For instance, Charles found that RA with a high disease activity score (DAS28 > 5.1) impaired CEC in a cross-sectional study [149]. With quantification by tandem mass spectrometry, Vivekanandan reported MPO-oxidation of HDL and reduced cholesterol efflux in RA [150]. Additionally, a meta-analysis of interventional studies showed that anti-rheumatic treatment ameliorated CEC independently of HDL-C [151]. In addition to RA, psoriasis, a chronic inflammatory skin disease, also increased risk of CVD as well as of stroke, and altered the HDL proteome and CEC [88]. Furthermore, impaired PON1 activity and elevated MPO levels were reported in systemic lupus erythematosus (SLE) [152]. Regarding inflammatory bowel disease (IBD), the antioxidative function of HDL was impaired in active Crohn’s disease [153]. However, because there have been few studies on the role of HDL function in autoimmune diseases, further research is needed.

### 6.5. Cancer

As a new aspect of HDL functionality, its pivotal role in cancer has been investigated in recent years [154]. For instance, ApoA-1 was found to exert an anti-tumorigenic effect on melanoma and ovarian cancer in vitro and vivo [155,156]. In addition, HDL inhibited cancer cell proliferation in prostate and breast cancer [157,158]. Furthermore, SR-B1 was shown to be upregulated in cancer cells [159], and PON1 activity was reduced in patients with cancer.

Although it is likely that HDL functionality, including cholesterol efflux and anti-oxidative functions, would affect the cancer microenvironment, the mechanism remains unclear, and the clinical application of HDL in anti-cancer therapy is a future issue to be addressed.

In recent years, the initiation of immune checkpoint inhibitors (ICI) has been associated with the risk of cardiovascular disease by modulating the inflammatory pathway [160,161]. Whether ICI affects HDL functionality and the anti-inflammatory function of HDL is expected to modulate ICI-mediated atherogenicity remains to be elucidated; however, further clinical studies are needed.

## 7. Conclusions and Future Perspective

As described in this review, research on HDL has a history of over 60 years and the time has come to focus on HDL functions rather than HDL-C concentration alone. Numerous clinical and basic studies have shown that HDL functions are impaired in various diseases and environments. Furthermore, HDL has become a potential therapeutic target for cardiovascular disease and cancer. In future research, a comprehensive understanding of HDL functionality will be essential for unveiling new aspects of HDL.

## Figures and Tables

**Figure 1 biomolecules-13-00105-f001:**
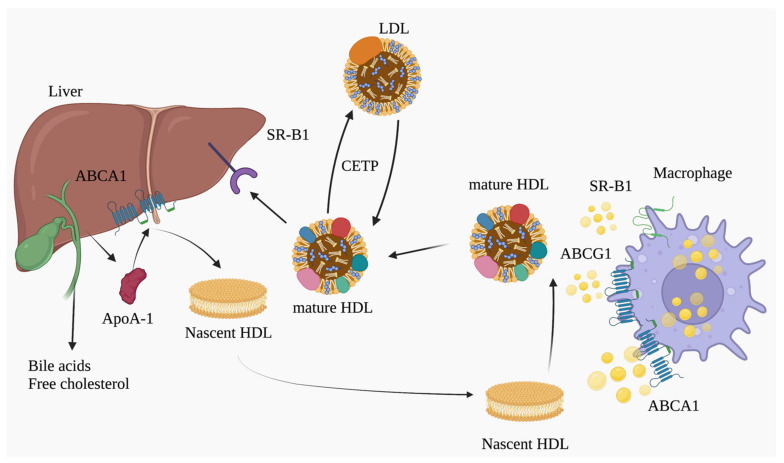
Schematic illustration of reverse cholesterol transport (RCT). RCT is a major atheroprotective function of HDL, a pathway by which HDL transports excess cholesterol from peripheral tissues, transfers it to the liver, and then excretes it from bile into the intestinal tract. Abbreviations: ABCA1: ATP-binding cassette protein A1, ABCG1: ATP-binding cassette protein G1, CETP: cholesteryl ester transfer protein, HDL: high-density lipoprotein, SR-B1: scavenger receptor class B member 1 (illustration was created using Biorender.com accessed on 7 December 2022).

**Table 1 biomolecules-13-00105-t001:** Summary of clinical studies on HDL functionality in cardiovascular disease.

Study	HDL Functionality	Design	Cohort	Main Findings
Khera A.V. et al., 2011 [33]	CEC	Cross-sectional	CAD: *n* = 442 Control: *n* = 351	OR for CAD per 1SD increase in CEC: 0.75 (0.63–0.90)
Ishikawa T. et al., 2015 [36]	CEC	Cross-sectional	CAD: *n* = 182 Control: *n* = 72	OR for CAD in CEC: 0.23 (0.056–0.91)
Ogura M. et al., 2016 [40]	CEC	Cross-sectional	Hetero FH with CVD: *n* = 76 without CVD: *n* = 151	OR for ASCVD per 1SD increase in CEC 0.95 (0.90–0.99)
Rohatgi A. et al., 2014 [34]	CEC	Prospective	the highest quartile of CEC (Q4) *n* = 731 the lowest quartile(Q1) *n* = 731	HR for ASCVD (Q4 vs. Q1) 0.33 (0.19–0.55)
Saleheen D. et al., 2015 [35]	CEC	Prospective	CAD: *n* = 1745, Control: *n* = 1749	OR for CAD per 1SD incresase in CEC (Top tertile of CEC vs. Bottom tertile) 0.80 (0.70–0.90)
Fujimoto D. et al., 2022 [38]	CUC	Cross-sectional	CAD with revascularization (*n* = 74)/non revaculariztion (*n* = 183)	OR for revascularization in CUC at the baseline 0.98 (0.969–1.000)
Ajala O.N. et al., 2020 [41]	HDL inflammatory index (HII)	Prospective	CAD: *n* = 517 Control: *n* = 517 HII: 0 to 0.5 *n* = 287 HII: >0.5 *n* = 720	HR for CVD and mortality (HII 0 to 0.5 vs. HII > 0.5) 1.53 (1.06–2.21)
Jia C. et al., 2021 [42]	HDL anti-inflammatory capacity	Cross-sectional	CAD: *n* = 340, Control: *n* = 340	OR for CVD per 1SD increase in anti-inflammatory capacity 0.74 (0.61–0.90)

Abbreviations: CAD: coronary artery disease, CEC: cholesterol efflux capacity, CUC: cholesterol uptake capacity, CUC: cholesterol uptake capacity, FH: familial hypercholesterolemia OR: odds ratio, HR: hazard ratio.

## Data Availability

Not applicable.

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
