# Peer review of "HDL Functions—Current Status and Future Perspectives"

_biomolecules, 2023, doi:10.3390/biom13010105_

Round 1
Reviewer 1 Report
General comments:
The authors have written a comprehensive review of HDL functions in the context of various translational science topics (diseases, exercise, nutrition). The review is interesting and informative. The authors appreciate the fact that concentration of an analyte does not indicate its kinetics or function. Therefore, they have focused upon HDL function (e.g., reverse cholesterol transport) rather than only focusing on the HDL-C concentration in plasma. This focus seems innovative and important. Suggestions below are aimed at identifying ways that the authors may improve the manuscript.
Major specific comments:
1) I do not see the Figure-1 image. Please provide it.
2) On page 3 the authors discuss CEC and CUC. This reviewer is not familiar with the differences between these assays, and readers may also be unfamiliar. Please explain in the text if they each measure a healthful process. Are they different methods of measuring the same thing? Or is efflux "good" and uptake "bad"?
3) Please provide more information about Table 1. For calculating HR or OR, what groups of patients were compared (e.g., groups with high vs low CEC?). This table needs additional information to be understandable.
Minor specific comments:
1) Change "heterogenicity" to "heterogeneity"
2) First paragraph of section 2.1: Is cholesterol excretion into the intestine is only through bile secretion? If so, please make that clear in the text. Also consider noting here that the liver can store cholesterol as cholesterol ester; that is another route for hepatic cholesterol use.
3) Line 100: Please note if this correlation is positive or is a negative/inverse correlation.
4) Reword the term "collaborating" in line 109.
5) Line 140-141: Please note if this association is positive or is inversely associated.
6) Line 230: Define "western diet". It is unclear what countries and ethnicities this refers too. And in each western country, most likely different cultures have different diets.
7) Line 264: 20kcal per kg body weight? That is a huge amount of exercise; that is noteworthy.
8) Section 5.1 and 5.3 has some paragraphs that are too short; this reviewer suggests to combine some of them together within each section.
9) For clarity, the following change might be useful at line 362: change "rather than HDL-C" to "rather than HDL-C concentration alone."
Author Response
.

Reviewer 2 Report
Manuscript titled " HDL functions - current status and future perspectives -" is a well performed review in the field of cardiovascular risk factors. Overall structure is of good quality, methods and paragraphs are well performed and clear to readers.
Authors should improve the manuscript in some parts:
1. in introduction, a proper description of environmental factors involved in low HDL levels should be sone. Fro example authors should describe how endocrine disruptors like bisphenol a are able to reduce HDL levels and to increase metabolic syndrome as well as cardiotoxicity of anticancer druds ( cite 10.1016/j.etap.2019.03.006 )
2. In discussion, authors should describe how cancer patients treated with Immune check point inhibitors have hgh risk of atherosclerosis and the patwhays involved ( nlrp3 inflamamsome and interleukin 1 and 6) ( cite 10.3390/jpm10040179) therefore these patients should be treated to increase HDL levels and reduce oxLDL levels. Please, discuss about this point.
Manuscript will be acceptable after minor revision.
Author Response
Reviewer2
We have the pleasure of submitting a revised manuscript that considers all the comments and suggestions by the reviewer. In this letter, we have included our responses to the reviewer’s critiques.
- in introduction, a proper description of environmental factors involved in low HDL levels should be sone. For example, authors should describe how endocrine disruptors like bisphenol a are able to reduce HDL levels and to increase metabolic syndrome as well as cardiotoxicity of anticancer drugs ( cite 10.1016/j.etap.2019.03.006 )
Response:
We appreciate the comments and suggestions.
As pointed out, plastic-associated chemicals including bisphenol A exerted atherogenicity and downregulated ApoA-1 expression. However, whether bisphenol A would affect HDL functionality remains unknown.
We have revised the manuscript accordingly and added the reference you suggested as follows:
Page 8, Lines 300-305
Recently, plastic-associated chemicals including bisphenol A and phthalates, have been the focus of research owing to their atherogenicity (119). In particular, BPA is associated with atherosclerosis (120-122) and exerts proinflammatory effects (123). Although it also downregulated ApoA-1 expression (124), the effect of BPA on HDL function remains to be elucidated and further studies are warranted.
- In discussion, authors should describe how cancer patients treated with Immune check point inhibitors have hgh risk of atherosclerosis and the patwhays involved ( nlrp3 inflamamsome and interleukin 1 and 6) ( cite 10.3390/jpm10040179) therefore these patients should be treated to increase HDL levels and reduce oxLDL levels. Please, discuss about this point.
Response:
Thank you for this important comment.
As pointed out, the immune checkpoint inhibitor (ICI) modulated inflammatory pathways and the initiation of ICI was associated with cardiovascular events in a clinical study (Circulation. 2020 Dec 15;142(24):2299-2311.) Whether ICI affects HDL functionality and the anti-inflammatory function of HDL is expected to modulate ICI-mediated atherogenicity remains to be elucidated; however, further clinical studies are needed.
We have revised our manuscript accordingly and added the reference you suggested.
Page 9, Lines 381-385
In recent years, the initiation of immune checkpoint inhibitors (ICI) have been associated with the risk of cardiovascular disease by modulating the inflammatory pathway (159, 160). Whether ICI affects HDL functionality and the anti-inflammatory function of HDL is expected to modulate ICI-mediated atherogenicity remains to be elucidated; however, further clinical studies are needed.
Round 2
Reviewer 1 Report
The authors have adequately responded to each suggestion from this reviewer.